# Evaluation of Thyroid Hormone Replacement Dosing in Morbidly Obese Hypothyroid Patients after Bariatric Surgery-Induced Weight Loss

**DOI:** 10.3390/jcm10163685

**Published:** 2021-08-19

**Authors:** Paula Juiz-Valiña, María Cordido, Elena Outeiriño-Blanco, Sonia Pértega, Paula Urones, María Jesús García-Brao, Enrique Mena, Lara Pena-Bello, Susana Sangiao-Alvarellos, Fernando Cordido

**Affiliations:** 1Grupo Fisiopatoloxía Endocrina, Nutricional e Médica (FENM), Facultad de Ciencias de la Salud, Universidade da Coruña, 15006 A Coruña, Spain; Paula.Juiz.Valina@sergas.es (P.J.-V.); Maria.Cordido.Carro@sergas.es (M.C.); Paula.Urones.Cuesta@sergas.es (P.U.); Maria.Lara.Pena.Bello@sergas.es (L.P.-B.); 2Instituto de Investigación Biomedica (INIBIC) and Centro de Investigaciones Científicas Avanzadas (CICA), Universidade da Coruña, 15006 A Coruña, Spain; 3Servicio Endocrinología y Nutrición, Hospital Universitario A Coruña, 15006 A Coruña, Spain; Elena.Outeirino.Blanco@sergas.es; 4Unidad Epidemiologia Clínica y Bioestadística, Hospital Universitario A Coruña, 15006 A Coruña, Spain; Sonia.Pertega.Diaz@sergas.es; 5Servicio Cirugía General y Digestiva, Hospital Universitario A Coruña, 15006 A Coruña, Spain; MA.Jesus.Garcia.Brao@sergas.es (M.J.G.-B.); Enrique.Mena.del.Rio@sergas.es (E.M.)

**Keywords:** obesity, bariatric surgery, hypothyroidism, thyroid hormone replacement

## Abstract

The most frequent endocrine disease in obese patients is hypothyroidism. To date, there are no clear data regarding what happens to the dose of levothyroxine (LT4) after bariatric surgery (BS). The objective of the present study was to evaluate thyroid hormone replacement dose in morbidly obese hypothyroid patients after BS-induced weight loss. We explore the best type of measured or estimated body weight for LT4 dosing. We performed an observational study evaluating patients with morbid obesity and hypothyroidism who underwent BS. We included 48 patients (three men). In morbidly obese hypothyroid patients 12 months after BS-induced weight loss, the total LT4 dose or the LT4 dose/kg ideal body weight did not change, while there was a significant increase in LT4 dose/body surface area, LT4 dose/kg weight, LT4 dose/kg adjusted body weight, LT4 dose/kg body fat, and LT4 dose/kg lean body weight. There were no differences in LT4 dose and its variation between sleeve gastrectomy (SG) and Roux-en-Y gastric bypass (RYGB). The present study strongly suggests that LT4 dosing in obese hypothyroid patients can be individually adapted more precisely if it is based on ideal body weight.

## 1. Introduction

The number of obese patients has continuously increased all over the world since 1980. Obesity is now considered as an emerging condition. The disease burden related to high body mass index (BMI) and obesity has increased since 1990 mainly due to cardiovascular disorders [1]. The prevalence of obesity in Spain in 2010 was 22.9% (24.4% among men and 21.4% among women) [2] and more than 20% in most European countries [3]. Moreover, the prevalence of obesity and serious obesity will continue to increase [4]. A weight loss of 5% improves metabolic performance in numerous organs concurrently, and progressive weight loss causes dose-dependent changes in key adipose tissue metabolic pathways [5,6]. Bariatric surgery (BS) compared with non-surgical obesity treatment has been shown to produce marked improvements in comorbidities and a higher decrease in mortality [7]. In obese patients, BS was associated with longer life expectancy than usual obesity management, although mortality remained higher in the obese groups than in the general population [8]. Apart from weight loss, a significant improvement in obesity-related comorbidities has also been reported after BS. These changes in patients occur in the early postoperative period, before weight loss appears, so the improvement may be mostly due to the hormonal changes induced with BS [9]. In contrast, in obese and type 2 diabetes patients treated with BS or diet, the metabolic benefits of BS and diet were similar and were apparently related to weight loss itself, with no obvious clinically important effects independent of weight loss [10].

The European Society of Endocrinology clinical practice guideline on the endocrine study in obesity recognizes the increased prevalence of many endocrine conditions in obese patients. The most frequent endocrine disease in obese patients is hypothyroidism. It is recommended to test all obese patients for thyroid function, due to the high prevalence of hypothyroidism in obesity [3]. Hypothyroidism, characterized by an elevated serum thyrotropin (thyroid-stimulating hormone, or TSH) level affects up to 10% of the adult people [11]. Excess body fat is associated with hormonal alterations, including diminished GH secretion [12,13,14,15] and thyroid alteration [14,16]. Body weight is routinely used to calculate the dose of levothyroxine (LT4) to administer to a patient with hypothyroidism [17]. Different reasons for increased requirements in obese subjects can be suggested: increased lean and fat mass [18], increased volume of distribution, or delayed gastrointestinal absorption [19]. Accordingly, it has been suggested that BS-induced weight loss may reduce the amount of LT4 needs [20,21].

By altering the normal anatomy and physiology of the gastrointestinal tract, the surgical procedure may lead to a decline in the absorption of LT4 and thus increase the needs of the drug [22]. At the present time, there are no clear data regarding what happens to the dose of LT4 after BS. Therefore, as Gadiraju et al. [23] suggest, studies are needed before making definitive conclusions. Moreover, the effect of different BS types on LT4 needs is unknown [24], and in particular, the different effect of malabsortive or restrictive types of BS is unclear.

Ojomo et al. [17] studied optimal thyroid dosing after thyroidectomy and concluded that the current weight-based thyroid replacement standard fails to appropriately dose underweight and overweight subjects [17]. A common formula to determine the initial dose of levothyroxine (LT4) for thyroid hormone replacement (THR) is to start the patient on 1.6 mcg/kg body weight in the postoperative period after total thyroidectomy without correction for the patient’s BMI [25]. This dose is subsequently titrated based on TSH values and symptomatology. Replacement dosage recommendations are often ambiguous in regard to whether the dose is based on measured body weight or estimated ideal body weight or estimated lean body weight (LBW). This distinction relates to the effect of obesity on dosage requirements. It has been shown that obese patients can develop thyrotoxicosis if dosing is based on the same calculations that were developed for the non-obese population [26], but no corrected dosing regimen has been uniformly recommended. The weight-based dosing of THR inappropriately overdoses overweight subjects. A more appropriate formula for THR adjustment should consider both the weight and BMI of the patient [25].

The objective of the present study was to evaluate THR dose in morbidly obese hypothyroid patients after BS-induced weight loss, in order to investigate the mechanism responsible for THR dosing in hypothyroid patients with morbid obesity. We studied predictors of LT4 variation after BS and also evaluate the effect of BS type. In addition, we explore the best type of measured or estimated body weight for THR dosing. This evaluation of the best type of measured or estimated body weight for THR dosing has not been studied before and after BS-induced weight loss. We hypothesized that THR dosing in patients with severe obesity and hypothyroidism is altered during BS-induced weight loss, and that weight loss induced by BS modifies the dose of LT4 in relation to their measured or estimated body weight.

## 2. Patients and Methods

### 2.1. Patients

We performed a retrospective observational study assessing patients with morbid obesity and hypothyroidism who underwent BS at the University Hospital of A Coruña, between January 2016 and December 2020. From our group of patients who underwent BS, we selected the patients diagnosed with hypothyroidism and being treated with LT4 (tablet formulation). We included 48 patients (3 men) in our study, with a median age of 47.2 years (39.2–52.3) (interquartile ranges). The inclusion criteria for BS were to be between 18 and 65 years old, have a BMI > 40 kg/m^2^ (or > 35 kg/m^2^ and at least one serious obesity-related health problem, such as high blood pressure, diabetes, or sleep apnea), failure of previous nonsurgical attempts at weight loss, and expectation that the patient would adhere to postoperative care and follow-up visits. Exclusion criteria were severe psychiatric illness, drug or alcohol abuse, lack of comprehension of benefits, risks, alternatives, expected outcomes, and lifestyle changes required with BS. A multidisciplinary team that includes a bariatric surgeon, an endocrinologist, and a psychiatrist evaluate all patients considered for BS, and based on the clinical characteristics (BMI, age, health problems) allocate the patient to sleeve gastrectomy (SG) or Roux-en-Y gastric bypass (RYGB). A total of 21 of the patients from our study (44%) were allocated to SG and 27 (56%) were allocated to RYGB. The study protocol was approved by our center’s ethics committee (Xunta de Galicia), approval code number: 2014/135, and written informed consent was obtained from all patients and controls. All of the studies were conducted in accordance with the Declaration of Helsinki.

### 2.2. Parameters Analyzed

The following data were analyzed: sex, age, BMI, body fat percentage, excessive weight loss in percentage, excessive BMI loss in percentage, TSH, free T4 (FT4), fasting glucose, fasting insulin, absolute LT4 dose, LT4 dose by patient actual weight, LT4 dose by patient ideal weight, LT4 dose by patient adjusted weight, LT4 dose by patient lean weight, LT4 dose by patient fat weight, LT4 dose by body surface area, and the type of BS performed (RYGB or SG). The data were assessed before BS and then at intervals of three, six and twelve months. All blood samples were collected after an overnight fast between 8:00 a.m. and 9:00 a.m. and immediately centrifuged, separated, and frozen at −80 °C. The primary endpoint was LT4 dose and LT4 dose in relation to their measured or estimated body weight. The secondary endpoint was the influence of the BS type.

### 2.3. Analytical Procedures

Serum TSH (mIU/L) and FT4 (ng/dL) levels were measured in serum from blood samples, by chemiluminescent immunoassay (ADVIA Centaur, Siemens, Erlangen, Germany) as previously published [27,28]. Serum insulin (µU/mL) was measured with a chemiluminescent immunometric assay (Immulite 2000 Insulin, DPC, Los Angeles, CA, USA) as previously published [27,28]. Glucose (mg/dL) was determined with an automatic glucose oxidase method (Roche Diagnostics, Mannheim, Germany). Total fat mass and lean body weight was calculated through bioelectrical impedance analysis (BIA). The BIA measurements were taken using a tetrapolar bioimpedantiometer BC-418 Segmental Body Composition Analyser (TANITA, Amsterdam, The Netherlands). The participants were examined while lightly dressed and barefoot, placing the feet on the metal footprints, grasping the hand grips with both hands, without moving and in a standing position. The measurement process was standard and was strictly supervised.

### 2.4. Calculations

Actual weight (AW) was the current weight of the patient. Ideal body weight (IBW) was determined as the predicted weight of a patient based on the patient’s height as derived by the Devine formula [29]. Lean body weight (LBW) was determined using BIA with a tetrapolar bioimpedantiometer BC-418 Segmental Body Composition Analyser (TANITA).

Adjusted body weight (ABW) was expressed as predicted body weight, which takes into account the IBW and AW to correct for the shortcomings of IBW, and it is considered more appropriate for use with patients with an actual body weight 30% greater than the predicted IBW. ABW is calculated based as IBW +0.4 (AW-IBW). Body surface area (BSA) was calculated as the common formula used for medication dose titrations [30].

Excessive BMI loss as a percentage was calculated using the formula: [(preoperative BMI-current BMI)/(preoperative BMI-25)] × 100.

### 2.5. Statistical Analysis

Descriptive analysis was used to determine the baseline characteristics of the patients included in the study. Continuous data are expressed as median and interquartile range (IR). Non-numerical variables are expressed as frequencies and percentages.

The Wilcoxon signed-rank test was used to compare the preoperative and 12-months post-surgery LT4 dose in the obese patients. Generalized estimating equations (GEE) models, with the autoregressive correlation structure, were used to evaluate the evolution over time of the LT4 dose and LT4 dose in relation to the measured or estimated body weight, adjusting for the type of BS. To assess the overall association between LT4 dosing and the variation in anthropometric parameters after BS, the repeated measures correlation was calculated [31].

For the statistical analyses, SPSS v24.0 and R v3.5.1 (IBM, Armnok, NY, USA) (with the *geepack* and *rmcorr* packages added) were used. Bilateral *p*-values of <0.05 were considered as statistically significant.

## 3. Results

### 3.1. Preoperative Characteristics

The preoperative characteristics of the morbidly obese hypothyroid patients (48 patients, 45 women and 3 men) are shown in Table 1.

The preoperative BMI (kg/m^2^; median, interquartile ranges) values were similar in the SG group and the RYGB group; 46.1 (40.6–51.0) vs. 47.6 (41.3–49.3) for the SG and RYGB group, respectively.

### 3.2. Evolution over Time of the Clinical and Analytical Parameters

The evolution over 3, 6, and 12 months after BS of the clinical and analytical parameters are shown in Table 2. No statistically significant differences were observed in TSH or free T4 values before and after BS, although there was a tendency to decrease in free T4 values after surgery.

In Figure 1 and Table 3, we show the evolution of the LT4 dose in hypothyroid patients before and after BS. Results from bivariate GEE models showed that the absolute LT4 dose did not vary significantly after BS nor did the LT4 dose/kg ideal body weight. On the contrary, there was a significant increase in LT4 dose/BSA (m^2^), LT4 dose/kg actual weight, LT4 dose/kg adjusted body weight, LT4 dose/kg body fat, or LT4 dose/kg lean body weight, as is also shown in Figure 1. The same results were obtained when comparing LT4 dose before and twelve months after BS (Table 4). In more than 75% of the patients, both the LT4 dose and the LT4 dose/kg ideal body weight they were receiving before BS were maintained 12 months after surgery. However, 12 months after BS, the LT4 dose/BSA (m^2^) the patients were receiving was on average 30.3% higher than before surgery. This average increase was 69.8% for LT4 dose/kg actual weight, 33.4% for LT4 dose/kg adjusted body weight, 178.7% for LT4 dose/kg body fat and 18.9% for LT4 dose/kg lean body weight.

No differences were observed in the LT4 dose and its variation after surgery between restrictive (SG) and malabsorptive (RYGB) techniques (Figure 2).

After adjusting for time and type of BS in bivariate GEE models, a significant trend for increase in LT4 dose/BSA values was determined (*p* < 0.001), estimating a mean increase around B = 0.88 units per month of follow-up. The same statiscally significant trend was identified for LT4 dose/kg actual weight (B = 0.03; *p* < 0.001), LT4 dose/kg adjusted body weight (B = 0.03; *p* < 0.001), LT4 dose/kg body fat (B = 0.18; *p* < 0.001), and LT4 dose/kg lean body weight (B = 0.02; *p* < 0.001). No significant changes in the follow-up was confirmed for LT4 dose (*p* = 0.142) nor for LT4 dose/kg ideal body weight (*p* = 0.095). Furthermore, the BS type was not significantly associated with LT4 dose changes except for LT4 dose/kg lean body weight, which was borderline statistically significant (Table 5).

In general, during the follow-up after BS, a greater BMI loss was significantly associated with an increase both in total LT4 dose and in LT4 dose adjusted to different weights, as it is deduced from the significant (*p* < 0.05) and negative correlation coefficients from Table 6. Variation in absolute LT4 dose and LT4 dose/kg ideal body weight was not significantly associated with the decrease in the percentage of body fat, in the kilograms of lean body weight lost, or with EWL and excessive BMI loss in percentage. However, a decrease in these anthropometric parameters was significantly correlated with an increase in LT4 dose/BSA, LT4 dose/kg actual weight, LT4 dose/kg adjusted body weight, and LT4 dose/kg body fat (Table 6).

## 4. Discussion

The principal result of the present study is that we have found that in morbidly obese hypothyroid patients 12 months after BS-induced weight loss, the total LT4 dose or the LT4 dose/kg ideal body weight did not change; nevertheless, there was a significant increase in LT4 dose/BSA, LT4 dose/kg weight, LT4 dose/kg adjusted body weight, LT4 dose/kg body fat, and LT4 dose/kg lean body weight. Variation in absolute LT4 dose and LT4 dose/kg ideal body weight was not significantly associated with the decrease in the percentage of body fat, in the kilograms of lean body weight lost, or with EWL and EBMIL. On the contrary, a decrease in these anthropometric parameters was significantly correlated with an increase in LT4 dose/BSA, LT4 dose/kg adjusted body weight, LT4 dose/kg actual weight, and LT4 dose/kg body fat. No differences were observed in LT4 dose and its variation after BS between SG and RYGB. For all we know, this is the first time that the best type of measured or estimated body weight for the THR dosing has been studied before and after BS-induced weight loss in morbidly obese hypothyroid patients.

The interference on LT4 absorption exerted by BS has been extensively studied, although with unclear results. With the advent of BS, especially with malabsorptive techniques, doubts about reduced drug and hormone absorption have been raised [32,33]. Hypothyroid patient treatment requires oral administration of LT4, and so there are concerns about its viability after BS [18]. After oral administration, approximately 60–80% of LT4 crosses the intestinal barrier. The LT4 absorption place is the jejunum and the upper part of the ileum [19]. Several factors may interfere with bowel LT4 absorption. Dietary components such as a fiber-enriched diet, malabsorption disorders, drugs that disrupt intestinal transport, and LT4 formulation (liquid vs. tablet formulation) are concurrent factors [34]. Whether or not BS should be included in this list remains controversial [23]. With regard to BS techniques involving gastric restriction such as SG, these are considered techniques that disturb drug absorption less than techniques involving intestinal diversion such as RYGB [35]. Despite an expected increased need for LT4 after BS, the data are controversial. A study comparing LT4 uptake before and after BS found no decrease in the absorption of the hormone, but only a retarded absorption of LT4 [18]. Various articles have reported an increased need for LT4 following jejunoileal bypass [36,37,38], suggesting the importance of a reduced absorptive surface [36,37,38]. These results are consistent with the increased TSH values observed in subjects treated with the same dose of T4 before and after BS [39,40]. On the contrary, other studies have described a reduction in LT4 requirements after RYGBP or SG [20,41]. In the study of Rudnicki et al. [42], BS improved thyroid function in hypothyroid obese patients. Zendel et al. [43] and Aggarwal et al. [41] have found that BS has a favorable impact on hypothyroid patients as seen by a reduction in LT4 dose. The study of Almunif et al. [44] has shown hypothyroidism improvement after SG [44]. In a recent meta-analysis, Azran et al. [45] have found that BS is associated with a decrease in total LT4 dose but with high heterogeneity between studies. In the article of Pedro et al. [24], there was no difference in the total LT4 dose twelve months after BS, although there was a significant increase in LT4 dose/kg actual weight. Fierabracci et al. [46] have found a decrease or no change in total LT4 dose but an increase of weight-based LT4 needs in patients after BS. Our results partially agree with the findings of the previous studies, as we have found that in morbidly obese hypothyroid patients, after BS-induced weight loss, the total LT4 or the LT4 dose/kg ideal body did not change, although the LT4 dose/BSA, LT4 dose/kg actual weight, LT4 dose/kg adjusted body weight, LT4 dose/kg body fat, and LT4 dose/kg lean body weight increased.

Julia et al. [47] found that weight-based LT4 dose increased in the RYGB group with no changes in the SG group two years after surgery. The authors conclude that RYGB and SG showed different changes in LT4 requirements. Conversely, and in agreement with Pedro et al. [24], we did not find BS type to be a predictor of either LT4 dose changes or its variation. Similarly, Rudnicki et al. [42] have found that SG and RYGB both improved thyroid function in hypothyroid obese patients and no procedure was superior [42]. This suggests that both procedures appear to have very similar effects on the hormone absorption. The mechanism by which these techniques interfere with the hormone absorption may be different although with very similar results. In RYGB, nutrient malabsorption and loss of gastric acidification may affect LT4 absorption. In restrictive techniques such as SG, the induced gastric alterations may be the main mechanism altering the hormone pharmacokinetics. Modified gastric emptying has been shown to interfere with LT4 absorption [48]. Since modified gastric emptying has been reported in all these techniques [49], this factor may at least partly explain the similar results between groups.

There are different potential mechanisms underlying why after BS-induced weight loss, the total LT4 dose did not change, although the relative LT4 dose (μg/kg) significantly increased. Two pharmacokinetic studies have been conducted to investigate T4 absorption [18,22]. Despite an improvement of absorption, Rubio et al. [18] perceived a significant retardation of LT4 absorption after BS. In the study of Gkotsina et al. [22], the pharmacokinetic parameters were similar before and after RYGB. However, Fallahi et al. [40], by using the same dose of LT4 before and after BS, observed that serum TSH increased after BS. Although the results in patients exclusively treated with malabsorptive techniques coincided in identifying an increase in the LT4 needs, results about LT4 requirements in patients treated with techniques combining restrictive and malabsorptive procedures are contradictory. This may depend on the different timing of LT4 ingestion and on the various effects that surgery may have on a patient’s gastrointestinal anatomy and physiology (modified gastric juice, dumping syndrome, modified gastric emptying, different microbial gut flora, variations of the lean and fat body mass ratio) [49,50]. A clear example of the drug effects is that in obese patients with diabetes and primary hypothyroidism on THR, metformin administration induces a fall in TSH that is probably due in part to metformin-induced weight loss [51]. In addition, the assessment of LT4 needs not always normalized by body weight does not allow for a proper and complete comparison of the results. Most of the studies with procedures combining restrictive and malabsorptive surgical techniques have found that BS is associated with a decrease in LT4 dosing [45]. However, the heterogeneity of these studies in patients who underwent BS with mixed restrictive and malabsorptive procedures does not make it possible to establish completely definitive conclusions about the net effect on LT4 need [52]. The decrease in lean body mass following BS could also potentially contribute to a decrease in LT4 dose [26]. After SG, despite a decrease in daily LT4 needs, correlating with weight loss, the absence of correlation with weight-adjusted dose suggests the involvement of confounding factors such as decreased LT4 absorption or altered thyroid function [21]. Rudnicki [42] found that BS improves thyroid function in hypothyroid obese patients, and no correlation was found between the percentage of weight loss and TSH decrease. This would suggest that the effect of BS on the improvement of thyroid function is due to mechanisms other than weight loss, probably hormonal changes [42]. In addition, altered LT4 pharmacokinetics in obese subjects have been described and attributed to altered plasma volume and delayed gastrointestinal absorption [46]. Based on these assumptions, a significant BS-induced weight loss would be expected to produce a reduction in LT4 needs. Another aspect to be considered is that hypothyroid treated patients could have metabolic differences with euthyroid subjects. Muraca et al. have found that obese hypothyroid patients in LT4 therapy, with a normal serum TSH level when compared with euthyroid subjects, are characterized by reduced resting energy expenditure, which is in accordance with the hypothesis that THR may not completely correct metabolic disturbances due to hypothyroidism [53]. Mele et al. [54] have found that compared with euthyroid obese patients, LT4 users presented higher adiposity. We have found that in morbidly obese hypothyroid patients after bariatric surgery-induced weight loss, the total LT4 dose did not change, but the relative LT4 dose (μg/kg actual weight) significantly increased. These data suggest that both the decreased LT4 absorption due to BS and decreased LTA need due to weight loss could contribute to THR change after BS.

In order to shorten the titration period, which can be long [25], different studies have considered the best way of dosing LT4 for patients with hypothyroidism and obesity. In the article of Papoian et al. [25], the weight-based dosing of THR overdoses overweight and obese subjects. Papoian et al. considered that a more appropriate formula for THR titration should include other aspects such as both the body weight and BMI of the subject, and they recommend using either the actual weight of the patients with adjustment of dosing based on the BMI or the adjusted BW without regard to the patient’s BMI [25]. Santini et al. have found that the THR dose to be administered to patients with hypothyroidism can be individually adapted more precisely if it is based on lean body mass [26]; these data may explain part of the results of the present study. LT4 needs depend on lean body mass, and an estimate of lean body mass could be helpful in shortening the time required to reach a stable LT4 dose, particularly in obese patients. In obese patients with hypothyroidism, the needs of LT4 are increased, due to an increase not only in fat body mass but also in lean body mass [30]. Moreover, Mele et al. have found that the LT4 dose was predicted by fat-free mass, hypothyroidism cause, and sex [54]. In the present study, we have found that in morbidly obese hypothyroid patients, after BS-induced weight loss, the LT4 dose/kg ideal body weight did not change, although there was a significant increase in LT4 dose/BSA, LT4 dose/kg weight, LT4 dose/kg adjusted body weight, LT4 dose/kg body fat, or LT4 dose/kg lean body weight. These data strongly suggest that THR dosing in obese hypothyroid patients can be individually adapted more precisely if it is based on ideal body weight.

We must recognize a number of limitations of the present study. A major limitation is the small sample size, so the study is unpowered to analyze the influence of different variables on the change in LT4 dose after BS. In particular, it cannot be ruled out that the fact that no statistically significant differences were detected between SG and RYGB may be due to a lack of statistical power. We did not consider certain variables that could influence the study, such as the concomitant use of other drugs, the heterogeneous nature of our obese patient group, with different comorbidities and treatments, and the fact that most of the patients were female. Furthermore, due to the retrospective nature of the study and its conduct in the usual clinical context, the follow-up of patients is not as exhaustive as in prospective studies. However, there are several strengths to our study. We evaluated THR dosing in morbidly obese hypothyroid patients at different time points after surgery, as most studies evaluated the change of TSH alone using only two moments (before and after surgery). Moreover, this is the first study to compare the LT4 dose per different types of body weight (AW, IBW, ABW, LBW, BSA) before and after BS.

## 5. Conclusions

In conclusion, the present study shows that in hypothyroid patients with morbid obesity, after BS-induced weight loss, the total LT4 dose or the LT4 dose/kg ideal body weight did not change, although there was a significant increase in LT4 dose/BSA, LT4 dose/kg actual weight, LT4 dose/kg adjusted body weight, LT4 dose/kg body fat, and LT4 dose/kg lean body weight. Variation in the absolute LT4 dose and the LT4 dose/kg ideal body weight was not significantly associated with the decrease in the percentage of weight lost. On the contrary, a decrease in these anthropometric parameters was significantly correlated with an increase in LT4 dose/BSA, LT4 dose/kg actual body weight, LT4 dose/kg adjusted body weight, and LT4 dose/kg body fat. No differences were observed in the LT4 dose and its variation after BS between SG and RYGB. The present study strongly suggests that THR dosing in obese hypothyroid patients can be individually adapted more precisely if it is based on ideal body weight.

## Figures and Tables

**Figure 1 jcm-10-03685-f001:**
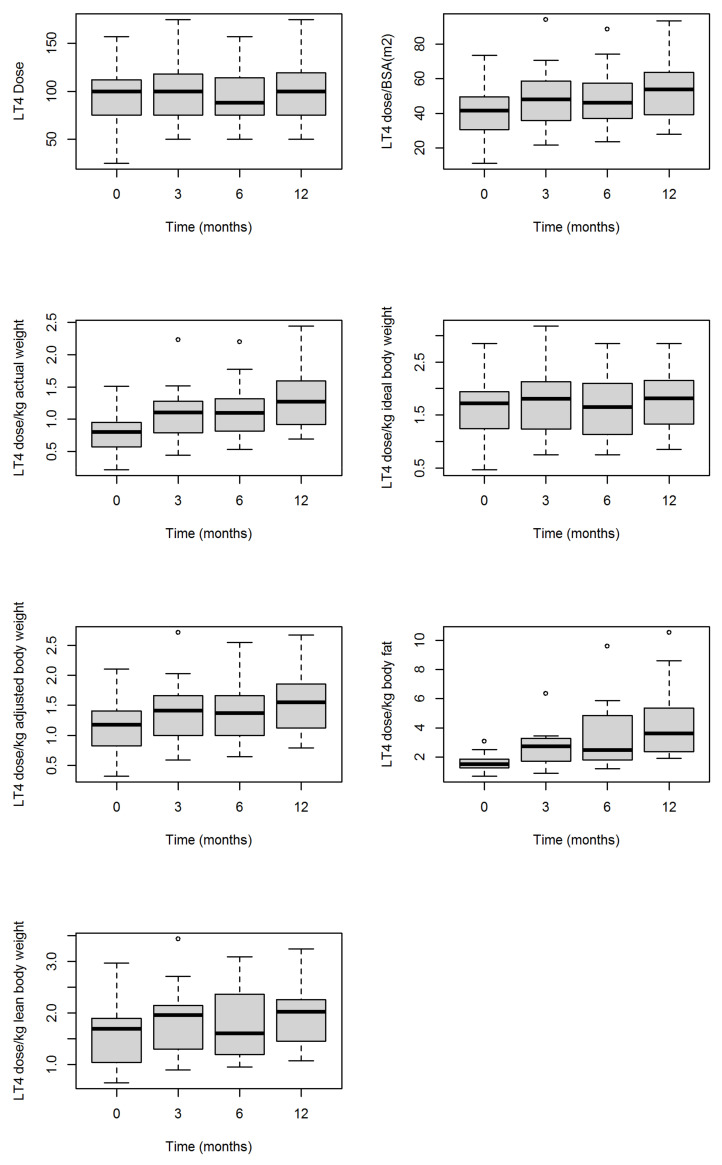
Levothyroxine (LT4) dose (median, interquartile ranges) change in hypothyroid patients before and 3, 6, and 12 months after bariatric surgery total and adjusted to different weights. BSA, body surface area. °, are outliers.

**Figure 2 jcm-10-03685-f002:**
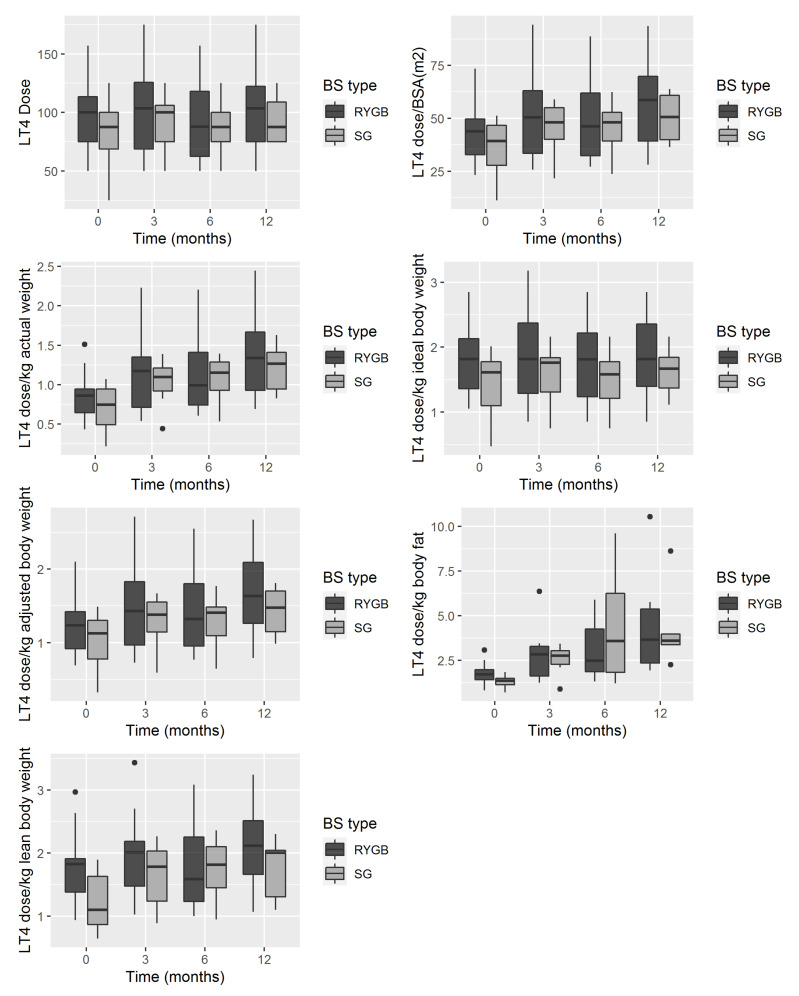
Levothyroxine (LT4) dose (median, interquartile ranges) change in hypothyroid patients after bariatric surgery (BS), according to BS type, sleeve gastrectomy (SG) or Roux-en-Y gastric bypass (RYGB). ●, are outliers.

**Table 1 jcm-10-03685-t001:** Preoperative characteristic of the morbidly obese hypothyroid patients (median, interquartile ranges).

	Median	25th Percentile	75th Percentile
BMI (kg/m^2^) t0	47.1	42.5	49.4
Body Fat (%) t0	51.2	49.1	53.3
Lean Body Weight (kg) t0	57.4	53.5	62.4
Body Surface Area (m^2^) t0	2.3	2.2	2.4
Fasting Glucose (mg/dL) t0	92.0	78.0	115.0
Fasting Insulin (µIU/mL) t0	6.4	3.6	11.5
Free T4 (ng/dL) t0	1.6	1.4	1.7
TSH (µU/mL) t0	2.6	1.2	6.9

BMI, body mass index; t0, time 0; TSH, thyroid stimulating hormone.

**Table 2 jcm-10-03685-t002:** Clinical and analytical parameters of the morbidly obese hypothyroid patients (median, interquartile ranges) three, six, and twelve months after surgery.

	3 Months after Surgery			6 Months after Surgery			12 Month safter Surgery		
	Median	P25	P75	Median	P25	P75	Median	P25	P75
BMI	38.0	33.6	40.2	32.8	29.6	36.5	31.6	29.1	34.0
EWL (%)	43.3	34.7	47.4	58.1	49.8	63.6	65.6	60.1	75.2
EBMIL (%)	48.2	37.0	52.8	64.8	53.4	76.2	71.5	65.6	80.1
Body Fat (%)	43.4	38.2	48.9	36.8	35.2	41.0	36.7	30.3	41.2
Body Surface Area (m^2^)	2.0	1.9	2.1	1.9	1.8	2.0	1.8	1.8	1.9
Fasting Glucose (mg/dL)	84.0	78.0	89.0	81.0	74.0	88.0	82.0	78.0	88.0
Fasting Insulin (µIU/mL)	5.7	3.6	6.0	5.2	3.4	9.2	7.8	3.1	10.0
Free T4 (ng/dL)	1.2	1.1	1.5	1.1	1.0	1.2	1.1	1.0	1.2
TSH (µU/mL)	2.1	1.1	3.2	3.1	2.0	3.7	3.3	1.7	5.0

P25, 25th percentile; P75, 75th percentile; BMI, body mass index; EWL, excessive weight loss in percentage; EBMIL, excessive BMI loss in percentage. TSH, thyroid stimulating hormone.

**Table 3 jcm-10-03685-t003:** Levothyroxine (LT4) dose (median, interquartile ranges) in hypothyroid patients before and after bariatric surgery.

	Before Surgery			3 Months after Surgery			6 Months after Surgery			12 Months after Surgery			
	Median	P25	P75	Median	P25	P75	Median	P25	P75	Median	P25	P75	*p* *
LT4 dose	100.0	75.0	112.0	100.0	75.0	122.0	88.0	75.0	114.0	100.0	75.0	119.5	0.058
LT4 dose/BSA (m^2^)	41.7	30.6	49.5	48.2	35.8	59.0	46.3	37.0	57.5	53.8	39.2	63.8	<0.001
LT4 dose/kg actual weight	0.8	0.6	0.9	1.1	0.8	1.3	1.1	0.8	1.3	1.3	0.9	1.6	<0.001
LT4 dose/kg ideal body weight	1.72	1.24	1.94	1.81	1.13	2.16	1.65	1.13	2.10	1.82	1.33	2.15	0.058
LT4 dose/kg adjusted body weight	1.2	0.8	1.4	1.4	1.0	1.7	1.4	1.0	1.7	1.6	1.1	1.9	<0.001
LT4 dose/kg body fat	1.5	1.3	1.9	2.8	1.7	3.3	2.5	1.8	4.8	3.6	2.4	5.4	<0.001
LT4 dose/kg lean body weight	1.7	1.0	1.9	2.0	1.1	2.2	1.6	1.2	2.4	2.0	1.5	2.3	<0.001

P25, 25th percentile; P75, 75th percentile; BSA, body surface area; * *p*-value from the generalized estimating equations (GEE) model.

**Table 4 jcm-10-03685-t004:** Absolute and relative levothyroxine (LT4) dose change in hypothyroid patients before and twelve months after bariatric surgery.

	Absolute Change at 12 Months			Relative Change at 12 Months (%)			
	Median	P25	P75	Median	P25	P75	*p* *
LT4 dose	0.00	0.00	0.00	0.00	0.00	0.00	0.083
LT4 dose/BSA (m^2^)	12.05	6.61	14.85	25.94	20.23	30.32	<0.001
LT4 dose/kg actual weight	0.53	0.29	0.63	58.62	44.56	69.84	<0.001
LT4 dose/kg ideal body weight	0.00	0.00	0.00	0.00	0.00	0.00	0.109
LT4 dose/kg adjusted body weight	0.33	0.22	0.51	26.35	21.43	33.38	<0.001
LT4 dose/kg body fat	2.02	1.59	3.23	134.86	113.13	178.67	0.001
LT4 dose/kg lean body weight	0.22	0.16	0.38	17.35	11.83	18.91	0.001

P25, 25th percentile; P75, 75th percentile; BSA, body surface area; * *p*-value from a paired Wilcoxon signed-rank test comparing Levothyroxine (LT4) dose before and 12 months after surge.

**Table 5 jcm-10-03685-t005:** Generalized estimating equation models examining change in levothyroxine (LT4) dose after bariatric surgery (BS), adjusting for BS type, sleeve gastrectomy (SG), or Roux-en-Y gastric bypass (RYGB).

	LT4 Dose			LT4 Dose/BSA (m^2^)		
	B	SE	*p*	B	SE	*p*
Intercept	100.16	9.06	<0.001	45.41	4.17	<0.001
Linear time (months after surgery)	0.65	0.20	0.142	0.88	5.13	<0.001
BS type (SG vs. RYGB)	−17.74	12.09	0.142	−9.07	5.74	0.110
	**LT4 dose/kg Ideal Body Weight**			**LT4 dose/kg Actual Weight**		
	**B**	**SE**	***p***	**B**	**SE**	***p***
Intercept	1.82	0.15	<0.001	0.90	0.09	<0.001
Linear time (months after surgery)	0.01	0.004	0.095	0.03	0.01	<0.001
BS type (SG vs. RYGB)	−0.40	0.20	0.051	−0.18	0.13	0.160
	**LT4 dose/kg Adjusted Body Weight**			**LT4 dose/kg Body Fat**		
Intercept	1.29	0.12	<0.001	1.87	0.24	<0.001
Linear time (months after surgery)	0.03	0.004	<0.001	0.18	0.04	<0.001
BS type (SG vs. RYGB)	−0.27	0.16	0.092	0.03	0.58	0.950
	**LT4 dose/kg L** **ean Body Weight**					
Intercept	1.82	0.17	<0.001			
Linear time (months after surgery)	0.02	0.004	<0.001			
BS type (SG vs. RYGB)	−0.47	0.23	0.041			

**Table 6 jcm-10-03685-t006:** Correlation between levothyroxine (LT4) dose change in hypothyroid patients and variation in anthropometric parameters after bariatric surgery. Values shown are repeated measures correlation coefficients together with their 95% confidence intervals.

	BMI	Body Fat (%)	Lean Body Weight (kg)	EWL (%)	EBMIL (%)
LT4 dose	−0.309 (−0.537; −0.040) *	−0.189 (−0.470; 0.125)	−0.232 (−0.500; 0.077)	−0.068 (−0.412; 0.294)	−0.087 (−0.428; 0.276)
LT4 dose/BSA (m^2^)	−0.855 (−0.915; −0.759) *	−0.817 (−0.899; −0.680) *	−0.784 (−0.879; −0.630) *	0.631 (0.356; 0.805) *	0.615 (0.333; 0.796) *
LT4 dose/kg actual Weight	−0.904 (−0.944; −0.838) *	−0.895 (−0.943; −0.811) *	−0.839 (−0.910; −0.718) *	0.851 (0.712; 0.926) *	0.841 (0.693; 0.921) *
LT4 dose/kg ideal body weight	−0.310 (−0.537; −0.040) *	−0.187 (−0.468; 0.128)	−0.226 (−0.496; 0.084)	−0.045 (−0.393; 0.314)	−0.062 (−0.408; 0.299)
LT4 dose/kg adjusted body weight	−0.863 (−0.920; −0.770) *	−0.814 (−0.897; −0.675) *	−0.783 (−0.878; −0.628) *	0.605 (0.313; 0.792) *	0.588 (0.289; 0.783) *
LT4 dose/kg body fat	−0.673 (−0.812; −0.461) *	−0.700 (−0.829; −0.500) *	−0.667 (−0.808; −0.452) *	0.496 (0.106; 0.754) *	0.495 (0.104; 0.753) *
LT4 dose/kg lean body weight	−0.730 (−0.849; −0.542) *	−0.590 (−0.762; −0.342) *	−0.647 (−0.798; −0.420) *	0.240 (−0.191; 0.594)	0.231 (−0.200; 0.588)

* *p*-value < 0.05 obtained from a repeated measures correlation analysis. A positive correlation coefficient indicates a positive relationship, with a decrease in a specific anthropometric parameter associated with a decrease in LT4 dose. Alternatively, a negative correlation coefficient indicates an inverse relationship, with a decrease in a specific anthropometric parameter associated with an increase in LT4 dose. BMI, body mass index; EWL, excessive weight loss in percentage; EBMIL, excessive BMI loss in percentage.

## Data Availability

All data analyzed during this study are included in this manuscript as well as in previously published articles cited in the references. If any information is needed, please contact the corresponding author.

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
