# Peer review of "Evaluation of Thyroid Hormone Replacement Dosing in Morbidly Obese Hypothyroid Patients after Bariatric Surgery-Induced Weight Loss"

_jcm, 2021, doi:10.3390/jcm10163685_

Round 1

Reviewer 1 Report

Valina et al. explored the effect of different bariatric surgery on weight loss and levothyroxine replacement therapy.  Although different previous studies aimed at define levothyroxine dose adjustment after bariatic procedure with conflicting results. Therefore, the issue is of potential interest for the audience.

However, I have some concerns that should be addressed, and I thus recommend that this paper undergoes major revisions.

  • In general, the manuscript is well written and intelligible, although the discussion is a bit long-winded. A throughout revision of this part of the manuscript would be almost beneficial in terms of clarity and readability.

2) The authors found no difference in levothyroxine dosing after bariatric surgery, without differences between RYGB and SG. This is in contrast with a recent metanalysis by Arzan C and collegues (DOI:10.1016/j.soard.2021.02.028) who found a significant decrease in levothyroxine dose following roux en Y gastric by pass or sleeve gastrectomy. I suggest that this data may deserve a comment throughout the discussion

3) Interestingly, the authors clearly shown how Levothyroxine adjusted for weight and lean body mass   was augmented after bariatric surgery, while no chenges were found in the absolute dose of levothyroxine. However, in the presented data a slight elevation of TSH is seen befor and after bariatric surgery, thus suggested an slight higher levothyroxine requirement.

 4) I would have appreciated a more detailed discussion over the emerging role of lean body mass as a major determinant of levothyroxine dosing, which may represent a possible explanation of the main results of this study. Indeed, it is well known that bariatric surgery determines substantial  changes in fat and lean body mass ratio.   In my opinion, this may deserve a more accurate dissertation.

5) although a bit long-winded,  I really appreciate the discussion on the multiple factors influencing levothyroxine  dosing. I suggest that the influence of different levothyroxine formulation (liquid vs tablet formulation) may also deserve to be quoted. I also invite the author to specify the formulation use in the present study.

Author Response

REPLY TO REVIEWERS

The changes and additional text are highlighted in red.

Reviewer 1:

Thank you for your comments. We appreciate the referee’s positive comments regarding our work.

1) We appreciate the reviewer’s comment regarding that in general, the manuscript is well written and intelligible, although the discussion is a bit long-winded. A throughout revision of this part of the manuscript would be almost beneficial in terms of clarity and readability.

Following the reviewer’s recommendations, in the new version we have made a throughout revision of the discussion to improve clarity and readability. Page 10, line 272. Page 10, line 274. Page 10, line 284. Page 11, line 312. Page 11, line 320. Page 11, line 328. Page 11, line 357. Page 12, line 377. Page 12, line 387.

2) We appreciate the reviewer’s comment regarding the article by Arzan C and colleagues.

Following the reviewer’s recommendations, in the new version we have included in the discussion a comment about the results of the metanalysis by Arzan C and colleagues [1]. Thank you for your comment. Page 10, line 292 and that paragraph.

3) We appreciate the reviewer’s comment about a slight elevation of TSH is seen before and after bariatric surgery. As you have correctly pointed out, the TSH elevation is very slight and there was not statistically significant difference in TSH values before and after BS. Furthermore, due to the retrospective nature of the study and its conduct in the usual clinical context, the follow-up of patients is not as exhaustive as in prospective studies.

Following your recommendation, we have included a comment in the limitations part of the study Thank you for your comment. Page 12, line 391.

4) We appreciate the reviewer’s comment about a more detailed discussion over the emerging role of lean body mass as a major determinant of levothyroxine dosing.

Following the reviewer’s recommendation, in the new version we have included a more detailed discussion over the emerging role of lean body mass as a major determinant of levothyroxine dosing, which may explain part of the results of the present study. Thank you for your comment. Page 12, line 370.

5) We appreciate the reviewer’s comment regarding the discussion on the multiple factors influencing levothyroxine dosing.

Following the reviewer’s recommendations, in the new version we have quoted in de discussion the influence of different levothyroxine formulation (liquid vs tablet formulation). Page 10, line 276. The formulation used in the present study was tablet formulation. Thank you for your comment. Page 3, line 106.

References

  1. Azran, C.; Hanhan-Shamshoum, N.; Irshied, T.; Ben-Shushan, T.; Dicker, D.; Dahan, A.; Matok, I. Hypothyroidism and levothyroxine therapy following bariatric surgery: a systematic review, meta-analysis, network meta-analysis, and meta-regression. Surg Obes Relat Dis 2021, 10.1016/j.soard.2021.02.028, doi:10.1016/j.soard.2021.02.028.

Reviewer 2 Report

The manuscript explores how threatment for hypothyroidism is affected after bariatric surgery. This is interesting because current dosage schemes are heavily weight based, and may not be suitable for obese patients or in patients with altered intestinal absorption.

Major points

I think the presentation of the data could have been more straight forward. The data basically shows how the patients remain at unaltered dosage of LT4 in spite of weight loss that «should have» rendered them hyperthyroid. Instead, their level of FT4 decreases markedly while TSH increases. This is not discussed by the authors and in my view under communicated the way the paper is written. I am missing information about what really happened to the LT4 dosage after surgery in individual patients, from what I see over 75 percent did not have their dosage adjusted at all. It may look like there was a lack of follow up, and that rising TSH values had no consequences for dosage. If this happened, the main message and conclusion of the paper is however still valid, it is more a matter of describing the results better.

The relationshiop between weight and LT4 requirement is well known in the general hypothyroid population. A reasonable interpretation of the current results is that the reduced requirement for LT4 after weight loss is caused by LT4 absorption problems after BS. The authors show how free T4 decreases from median 1.6 ng/L to 1.1 ng/L after surgery. In my opinion, the authors show that patients develop relative hypothyroidism after BS and weight loss, which was not compensated. It is interesting that the weight loss did not have the opposite effect, and the only reasonable explanation would be malabsorption induced by the BS. Again, it would be interesting to know more details about individual data. A lot of individual differences could still be masked the way the data are presented. For example, did all patients have a slight drop it free T4 levels from time 0 to 3 months postop? If so, malabsorption is very likely.

The aim of the study is not well formulated and hard to understand in both the abstract and main text. I think that this is mostly a language issue, as there are numerous imprecise formulations throughout the text. For example line 294: the potential mechanisms do no underly THE REASON, but the effect itself.

I find that the analysis in Table 5 and 6 are hard to understand not explained well enough in the text.

Minor points

Abbreviations in abstract lack outspelling: SG and RYGB.

I am not sure that you can conclude that there is no difference in LT4 absorption between SG and RYGB, the study is underpowered to answer this. But the results indeed suggests that this is the case.

A comparison of basleline characteristics in the two groups SG and RYGB would be adequate.

Author Response

REPLY TO REVIEWERS

The changes and additional text are highlighted in red.

Reviewer 2:

Thank you for your comments. We appreciate the referee’s positive comments regarding our work.

Major points

-We appreciate the reviewer’s comment regarding that the presentation of the data could have been more straight forward. But we try to present the results in order to answer the main objective of the study. Namely, the aim of this study was to evaluate thyroid hormone replacement (THR) dose in morbidly obese hypothyroid patients after BS-induced weight loss, in order to investigate the mechanism responsible for THR dosing in hypothyroid patients with morbid obesity. No statistically significant differences were observed in TSH or free T4 values before and after BS, although there was a tendency to decrease in free T4 values after surgery. Furthermore, due to the retrospective nature of the study and its conduct in the usual clinical context, the follow-up of patients is not as exhaustive as in prospective studies. The LT4 dose before and after surgery was adjusted individually during their usual clinical follow-up visits, and some of the patients due to non-compliance, lack of adequate follow-up or other reasons did not have the LT4 dosage perfectly adjusted both before and after surgery. Moreover, as you well know, free T4 levels are relatively insensitive to small alterations of thyroxine dosage [1]. But, as you have correctly pointed out, we fully agree that the main message of the paper is still valid.

Following your recommendation, in the new version we have better described the results and we have also included a comment in the limitations part of the study. Thank you for your comment. Page 5, line 188. Page 8, line 224. Page 9, line 249. Page 12, line 391.

-We appreciate the reviewer’s comment regarding that a reasonable interpretation of the current results is that the reduced requirement for LT4 after weight loss is caused by LT4 absorption problems after BS. In the reviewer’s opinion, the authors show that patients develop relative hypothyroidism after BS and weight loss, which was not compensated. It is interesting that the weight loss did not have the opposite effect, and the only reasonable explanation would be malabsorption induced by the BS. It is important to point out that there was not statistically significant difference in TSH or free T4 values before and after BS, although there was a tendency to decrease in free T4 values after surgery. Not all patients had a slight drop in free T4 levels from time 0 to 3 months postoperative. Furthermore, as you well know, free T4 levels are relatively insensitive to small alterations of thyroxine dosage, in contrast to TSH measurements [1]. We completely agree with most of your comments as these data suggest that both the decreased LT4 absorption due to BS and decreased LT4 need due to weight loss could contribute to thyroid hormone replacement change after BS (in Discussion, page 12, line 361).

Following your recommendation, in the new version we have better described the results and have also included a comment in the limitations part of the study. Thank you for your comment. Page 4, line 188. Page 9, line 239. Page 12, line 361. Page 12, line 391.

-We appreciate the reviewer’s comment regarding that the aim of the study is not well formulated and hard to understand in both the abstract and main text.

Following the reviewer’s recommendations, in the new version we have clarified this aspect and corrected the aim of the study. We try to be more precise throughout the text. We have also corrected the text on line 294 (now 316). Thank you for your comment. Page 1, line 19. Page 2, line 90. Page 11, line 316.

-We appreciate the reviewer’s comment regarding that the analysis in Table 5 and 6 are hard to understand not explained well enough in the text.

Following the reviewer’s suggestion, in the new version we have modified the text in the manuscript, in order to clarify the results shown both in Table 5 and 6.  Thank you for your comment. Page 8, line 224. Page 9, line 239. Page 9, line 249.

Minor points

-Following the reviewer’s recommendations, we have included in abstract the out spelling for SG and RYGB. Thank you for your comment. Page 1, line 28.

-As the reviewer pointed out, a major limitation of this study is the small sample size, so it might be unpowered to detect differences between groups or to analyze the influence of different variables on the change in levothyroxine dose after bariatric surgery. In particular, it cannot be ruled out that the fact that no statistically significant differences were detected between SG and RYGB may be due to the lack of statistical power. Furthermore, this is a controversial topic in the literature [2-4].

In order to address this issue, several modifications were made in the manuscript in the description of the results regarding the type of BS. Furthermore, a comment on this point was added in the discussion section, recognizing it as a major limitation. Page 8, line 231. Page 10, line 264. Page 12, line 407. Page 12, line 410.

-We appreciate the reviewer’s comment regarding that a comparison of baseline characteristics in the two groups SG and RYGB would be adequate. Due to the fact that in the opinion of the reviewer this is a minor point and the presence of 6 tables and 2 figures in the article, and the length of the work, we have included the most important points.  

Following the reviewer’s recommendations, in the new version we have included some of the baseline characteristics of the two groups SG and RYGB. Thank you for your comment. Page 4, line 179.

References

  1. Carr, D.; McLeod, D.T.; Parry, G.; Thornes, H.M. Fine adjustment of thyroxine replacement dosage: comparison of the thyrotrophin releasing hormone test using a sensitive thyrotrophin assay with measurement of free thyroid hormones and clinical assessment. Clin Endocrinol (Oxf) 1988, 28, 325-333, doi:10.1111/j.1365-2265.1988.tb01219.x.
  2. Pedro, J.; Cunha, F.; Souteiro, P.; Neves, J.S.; Guerreiro, V.; Magalhaes, D.; Bettencourt-Silva, R.; Oliveira, S.C.; Costa, M.M.; Queiros, J., et al. The Effect of the Bariatric Surgery Type on the Levothyroxine Dose of Morbidly Obese Hypothyroid Patients. Obes Surg 2018, 28, 3538-3543, doi:10.1007/s11695-018-3388-410.1007/s11695-018-3388-4 [pii].
  3. Rudnicki, Y.; Slavin, M.; Keidar, A.; Kent, I.; Berkovich, L.; Tiomkin, V.; Inbar, R.; Avital, S. The effect of bariatric surgery on hypothyroidism: Sleeve gastrectomy versus gastric bypass. Surg Obes Relat Dis 2018, 14, 1297-1303, doi:10.1016/j.soard.2018.06.008.
  4. Julia, H.; Benaiges, D.; Molla, P.; Pedro-Botet, J.; Villatoro, M.; Fontane, L.; Ramon, J.M.; Climent, E.; Flores Le Roux, J.A.; Goday, A. Changes in Thyroid Replacement Therapy after Bariatric Surgery: Differences between Laparoscopic Roux-en-Y Gastric Bypass and Laparoscopic Sleeve Gastrectomy. Obes Surg 2019, 29, 2593-2599, doi:10.1007/s11695-019-03890-9.

Round 2

Reviewer 2 Report

The manuscript has been improved after the first version. However, there are still several typos in the manuscript, a few examples:

Line 43: Ref 5 and 6 should be joined wihtin the same brackets

Line 44: «more marked», consider removing one of the words

Line 55: Capital S in Society

Line 61: punctuation before Ref 11 should be removed, also double spacing following

Line 258-259: space before «/kg» should be removed

Line 268: LT4 instendig of T4?

My suggestion to show individual data has not been followed. Perhaps not such a big issue, but it would allow the interested reader to explore the data further in search of mechanisms.

I think you need to show the development of TSH and FT4 after surgery for individual groups, SG vs RYGB, in order to support your last conclusion in the abstract and the last conclusion in line 410, that the two methods did not affect absroption differently. In line 179-181 you mention that the groups were similar for BMI, but what about other preop parameters?

Line 229: suggest removing «On the contrary»

Line 239: suggest adding, «especially regarding LT4 dosage adjustment after surgery».

Author Response

Reviewer 2

The changes and additional text are highlighted in red

Thank you for your comments. We appreciate the referee’s positive comments regarding our work.

Following the reviewer’s recommendations, in the new version we have corrected the typos throughout the text. Thank you for your comment. Lines 43, 44, 55, 61, 258-259, 268

Following the reviewer’s recommendations, in the new version we have withdrawn our conclusion in the abstract and the Discussion part of the article. Thank you for your comment. Now abstract and line 407.

Following the reviewer’s recommendations, in the new version we have included some of your suggestions. Thank you for your comment. Line 229.